# Assessment of Triboelectric Nanogenerators for Electric Field Energy Harvesting

**DOI:** 10.3390/s24082507

**Published:** 2024-04-14

**Authors:** Oswaldo Menéndez, Juan Villacrés, Alvaro Prado, Juan P. Vásconez, Fernando Auat-Cheein

**Affiliations:** 1Departamento de Ingeniería de Sistemas y Computación, Universidad Católica del Norte, Antofagasta 1249004, Chile; alvaro.prado@ucn.cl; 2Department of Biological and Agricultural Engineering, University of California, Davis, CA 95616-8678, USA; jvillacres@ucdavis.edu; 3Energy Transformation Center, Faculty of Engineering, Universidad Andrés Bello, Santiago 7500000, Chile; juan.vasconez@unab.cl; 4Department of Electronic Engineering, Federico Santa María Technical University, Valparaíso 2390123, Chile; f.auat@hw.ac.uk; 5School of Engineering and Physical Sciences, Heriot-Watt University, Edinburgh EH14 4AS, UK

**Keywords:** energy harvesting, electromagnetic induction, electromagnetic coupling, capacitance transducers, nanogenerators

## Abstract

Electric-field energy harvesters (EFEHs) have emerged as a promising technology for harnessing the electric field surrounding energized environments. Current research indicates that EFEHs are closely associated with Tribo-Electric Nano-Generators (TENGs). However, the performance of TENGs in energized environments remains unclear. This work aims to evaluate the performance of TENGs in electric-field energy harvesting applications. For this purpose, TENGs of different sizes, operating in single-electrode mode were conceptualized, assembled, and experimentally tested. Each TENG was mounted on a 1.5 HP single-phase induction motor, operating at nominal parameters of 8 A, 230 V, and 50 Hz. In addition, the contact layer was mounted on a linear motor to control kinematic stimuli. The TENGs successfully induced electric fields and provided satisfactory performance to collect electrostatic charges in fairly variable electric fields. Experimental findings disclosed an approximate increase in energy collection ranging from 1.51% to 10.49% when utilizing TENGs compared to simple EFEHs. The observed correlation between power density and electric field highlights TENGs as a more efficient energy source in electrified environments compared to EFEHs, thereby contributing to the ongoing research objectives of the authors.

## 1. Introduction

Nowadays, the Internet of Energy (IoE) serves as a quintessential component in today’s industry, enabling precise and efficient bidirectional communication among diverse power system assets. The IoE technologies enhance the coordination and communication between energy sources, storage systems, and consumption devices, contributing to the overall effectiveness and reliability of the power infrastructure [1,2]. In general, the IoE paradigm comprises billions of intelligent Wireless Sensor Nodes (WSNs), which measure different variables related to the power system behavior [3]. Nevertheless, additional information in the literature highlights concerns associated with large-scale IoE deployments within Wireless Sensor Networks [4]. For instance, wireless networked sensors pose issues such as elevated power consumption across a large set of sensors, ongoing maintenance limitations, and a shortened lifespan [5]. Thus, there is a growing interest in exploring new methods for battery-less sensors. In this context, energy harvesting (EH) technologies emerge as a potential alternative to complement or even replace batteries powering WSNs [6,7].

Triboelectric nanogenerators (TENGs) are EH devices that transform mechanical stimuli into electrical signals. Recently, TENGs have been widely used in industrial applications due to their adaptability to various energy sources, resistance to external disturbances, flexibility in rugged environments, and high power density [1,8]. Among the most relevant applications are measuring ambient variables (e.g., humidity, pressure, temperature), detecting chemical compounds (e.g., NO_x_, CO, and CO_2_), and sensing physical parameters (e.g., oscillations, vibrations, acceleration levels) [9,10,11,12,13,14,15,16,17]. The success of TENGs with a large number of available mechanical sources to generate energy in power systems influences the introduction and development of TENGs deployments in this field.

Different prototypes of TENGs capable of capturing energy from magnetic fields in domestic environments have recently been reported in the literature [1]. For example, in [18], the authors presented a designed TENG composed of a cantilever attached to a magnet piece, which was fixed at one extreme of the test structure to prevent power field leakage. The electric setup was employed to power LED arrays and low-power sensors, providing approximately 4.8 mW. A similar multi-scale surface magneto TENG apparatus has also been developed by Kwak et al. [19], where battery-less temperature sensors were powered by this hybrid technology. Similar to the former electromechanical layout, A TENG assembled with multiferroic materials was presented in [20], resulting in better capture of electrical power generation when combining triboelectric and piezoelectric effects. On the other hand, the success of low-voltage applications has also enabled the introduction of TENGs in power systems [1]. A detailed study by Z. Yuan et al. (2021) involved the development of a swinging TENG that vibrates due to interactions between external magnet fields and the magnetic field produced by electric power lines [21]. In addition, TENGs have also been used to capture electric power from vibrations, mechanical stimuli, and the rotation of electrical engines [8,22,23,24].

General schemes of Electric Field Energy Harvesters (EFEHs) are often built upon two conductive electrodes mounted near voltage sources, such as domestic wiring, power lines, substations, or power generation stations [1]. Several authors have introduced dielectric materials between electrodes to achieve multi-layer developments when it is required to harness dispersed energy in conservative-scale devices, enabling the accumulation of generated energy. This approach also results in the reduction in the impedance values for the energy harvester, consequently increasing the output power [6]. Most previous studies on EFEHs are based on high-voltage transmission lines because of large, solid, and robust surrounding electric fields [25,26,27]. However, the need to develop energy harvesters for ultra-low-power sensors, microchips, or radio transceivers, as well as the success of EFEH technologies in high-voltage settings has resulted in EFEH devices generally adopted in low-voltage power systems [6,28,29,30]. It is worth noting that TENGs and EFEHs usually share complementary structural layouts [1]. Thus, TENGs might be used as a hybrid device to collect electrical energy from mechanical stimuli and electric fields existing in electrified environments.

Although the recent evidence suggests a direct structural relation between TENGs and EFEHs, particularly in the basic design [1], their performance and efficiency in the energy harvesting process using the induced electric field by the combination of electric field and vibrations have not yet been formally studied. To address this previously unaccounted phenomenon, different-length TENGs were assembled, analyzed, and experimentally tested. Indeed, to exploit its surrounding electric field and mechanical oscillations, each energy harvester was directly mounted on the cover of a 1.5 HP single-phase induction motor, operating at nominal parameters of 8 A, 230 VAC, and 50 Hz. In addition, a functional prototype based on the best performance analyzed harvester is also proposed. Finally, several practical applications are presented as study matters.

The remainder of this work is organized as follows. Section 2 reviews the characteristics of the EF-TENG and presents the EF-TENG design. In Section 3, a simulation for TENG and EFEH is conducted to assess their electrical performance in controlled conditions. Section 4 assessed the energy harvester performance for a low-voltage prototype. The main applications of the proposed technology are analyzed in Section 5. Here, several applications in the field are presented. Finally, Section 6 details the conclusions of this work.

## 2. Material and Methods

This section presents the materials and methods necessary for replicating the energy harvester design and conducting experimental tests. Additionally, we evaluated the functional concept and key technical characteristics of the proposed TENGs. Each harvester is studied based on tests of open-circuit voltage and short-circuit current.

### 2.1. Hardware Design of the Energy Harvester

As Figure 1 illustrates, the proposed TENG is composed of three main layers: a collection plate and two dielectric materials with opposite electrostatic affinity. A 100 μm thickness aluminum (Al) foil was selected as the collection plate because of the low cost of the material. Although the charge collection of Al layers is lower than that of other conductive materials (e.g., copper) [6], the Al layers have several attractive features such as high material flexibility, resistance to corrosion, lightweight, and cost-effectiveness. The aluminum layer was cleaned with isopropyl alcohol to remove non-conductive materials and impurities. Subsequently, it was dried at 20 °C for 1 h. To ensure volume uniformity, the aluminum layer was cut into regularly sized pieces, each 3.5 cm in width and varying in length based on the breadth of the motor cover. For analysis of the harvested energy distribution, ten Al layers ranging from 1 cm to 10 cm in steps of 1 cm were considered. 

A thin glue layer was scattered onto the Al layer, ensuring that the Al layer was covered. Accordingly, a 200 μm-thick layer of polylactic acid (PLA) was directly deposited onto the glue layer using a Creality Ender-Pro 3 3D printer (Creality, Valparaíso, Chile). The printing accuracy was ±100 μm according to the maximum quality profile, reducing the size of micro-structures in the layers and increasing the contact surface. Additional printing parameters were determined based on prior research about the fabrication of dielectric layers using 3D printing technology [31]. The remainder of this study refers to the entire structure formed by the Al and PLA layers as the fixed layer. The fixed layer (Al face) was directly mounted on a 300 μm thick PLA surface, which was placed on the single-phase motor chassis (1.5 HP, 8 A, 230 V, 50 Hz). For analysis of the harvested energy, the power voltage supply and motor speed are controlled using a 2 KVA single-phase voltage regulator. 

To design a reliable and robust harvesting system, the material’s ability to gain or lose electrons is required, as detailed in the triboelectric series [32]. Recent evidence suggests that cellulose acetate (CA) exhibits high performance and reliability in terms of electron conductivity, along with being cost-effective [33]. In this context, a 0.15 mm thick cellulose acetate foil was selected as the second dielectric layer. Due to cost-effectiveness and availability considerations, the TENG was assembled with PLA and CA. Additionally, PLA was selected for developing a printed TENG using low-cost technologies like 3D printers. PLA has been previously reported as a positive triboelectric material [34,35]. In contrast, Wang et al. demonstrated that CA exhibits a notably high negative charge affinity, surpassing even that of PTFE [36,37]. The layer size is defined following the same previous methodology described for the electrodes. For reproducibility, the experiments were performed in a controlled environment, i.e., it was necessary to control the voltage and oscillation level of the motor. While the system was designed to harness vibrations and the electric field generated in motors, controlling the vibration level reliably poses challenges. Therefore, the CA layer was mounted on a mobile structure connected to a linear motor using double-sided tape to simulate motion. It used a Creality Ender-Pro 3 3D printer to build the mobile structure.

Displacement current-based harvesters face two significant constraints: (i) low-power density and (ii) high capacitive impedance [1,38]. To address these challenges, the assumptions made in [6,28,39] were adopted, where the authors introduced rectification and storage stages as prerequisites for the proper operation of the energy harvester. According to [29,40], the power density of harvested energy is higher when employing full-wave rectifiers compared to half-wave rectifiers in the electric-field energy harvesting process. Although this behavior could be considered valid for EFEHs, this is not the case for TENGs, where half-wave circuits provide a better performance in terms of collected energy [41]. As the electric field is the primary energy source in this work, the wave rectifier stage consists of a DB-107 full-bridge rectifier (Comchip, Valparaíso, Chile). Here, the chip has a reverse voltage of 1000 V, a voltage drop of 1.1 V, and a junction capacitance of 25 pF. The alternating current pins were connected to the harvesting plate (Al layer), referring to the ground. Additionally, an important parameter to consider is the traffic rate in the data transmission between the sensor and ground station [42]. In this context, the selection of a 4.7 μF metalized polypropylene film capacitor reduces the charging time and increases the gathered energy. According to the full bridge rectifier manufacturer, the leakage current is approximately 22 nA for a voltage of 103 V. Table 1 summarizes the main technical characteristics of the hardware regarding sensors and instruments used in this study.

### 2.2. Sensing System

The methodology described in this section adapts the guidelines of Shao et al. [43]. The operating principle of the proposed TENG can be summarized in four stages, as shown in Figure 2. As Figure 2a illustrates, the proposed method assumes, in principle, that the TENG is not under external oscillations, vibrations, or motion. Actually, the harvester does not generate electrons during this step due to the triboelectric effect. However, the electrostatic induction produces charge accumulation in the harvesting plate because of the surrounding electric field. To simulate realistic experimental conditions, an external linear motor compresses the TENG, increasing the contact area and generating an electric potential, as shown in Figure 2c. Therefore, an alternating current flows from the harvester through a grounded external load. When the TENG achieves the maximum compressed condition, the electric potential reaches the maximum value. Finally, the voltage returns to zero when the TENG is again decompressed, as shown in Figure 2d.

### 2.3. Data Processing

Figure 1 depicts the suggested harvester’s architecture. Each stage in Figure 1 is explained as follows:To ensure reproducible results, the experiments should be performed in a controlled environment. In this regard, experimental tests of the harvesters were developed by fixing the laboratory temperature to standard temperature levels according to the National Institute of Standards and Technology, i.e., 20 °C. In addition, relative humidity was set to about 50%. Each harvest sample was registered by taking a full-HD color image with a 16-megapixel monocular camera (Logitech, Santiago de Chile, Chile) within a validated digit ranging from 0 to 9.The characterization of the TENG was conducted by interfacing the harvester with various resistors, and the resulting voltage was systematically measured. Voltage measurements were carried out using a Keithley 6514 electrometer (Keithley, Tektronix, Santiago de Chile, Chile) connected to an industrial computer. The electrometer, configured with an IEEE-488 bus provider (refer to the Keithley 6514 user manual for details), facilitated the acquisition of up to 500 sensor readings per second. To enhance precision and mitigate the impact of uncertainties, a series of ten measurements were performed. Subsequently, the computed values were determined by averaging the acquired sensor database.To investigate the effects of electric field and oscillations in the energy harvester we evaluated the harvest performance in three different scenarios: (i) an EFEH, (ii) a TENG, and (iii) combining the former methods gathered in a hybrid device. Initially, the performance of the TENG as an EFEH was assessed. For this purpose, the linear motor was temporarily turned off. The second mode follows the working mechanism described in [43], i.e., the TENG operates in single-electrode mode. Finally, the third mode analyzed the hybrid behavior of the TENG, following guidelines described in Section 2.2.When running the trials, a critical thing to bear in mind is the practical harvester’s application. In this context, we investigate the performance of the harvesters in low-duty cycle applications. We have used the experience learned from previous experimental work to determine that the maximum frequency for reporting the status of smart-city assets is 30 min [28,42]. Here, the charging profile of a 4.7 μF capacitor was acquired using a Keithley-6514 electrometer. The baud rate was 9600 bps since it is unnecessary to use a high sampling time.

## 3. Simulation Results

This section evaluates the electrical performance of the proposed TENG in a simulated environment. The proposed device is assessed based on the open-circuit voltage ($V_{\mathrm{OC}}$) and short-circuit charge ($Q_{\mathrm{SC}}$). All experiments were designed and conducted using COMSOL Multiphysics 5.4 and MATLAB 2023a. To reduce the computational cost, we investigate the performance of a TENG with a reduced-area harvester. The dimensions of the TENG area range from 2 × 2 mm^2^ to 10 × 10 mm^2^. The main objective is to validate that a TENG surpasses an EFEH in charge collection within an electrified environment. Because of the high computational demands of the finite element method, we conducted all simulations on a high-performance computer system powered by an Intel(R) Core(TM) i7-10700F processor (Intel, Santiago de Chile, Chile) clocked at 2.90 GHz, boasting eight cores. Furthermore, the system is equipped with an NVIDIA GeForce RTX 3070 GPU and possesses 16 GB of VRAM (NVIDIA, Santiago de Chile, Chile).

As depicted in Figure 3, the proposed TENG demonstrates functionality across three distinct operational modes described in Section 2.3. Figure 3a–f details the electrical characteristics of the TENG when employed as an EFEH. The $V_{\mathrm{OC}}$ remains unaffected by variations in electrode size, as shown in Figure 3a. In contrast, the $Q_{\mathrm{SC}}$ exhibits a proportional dependence on the electrode size, as shown in Figure 3b. Moreover, it is evident that the electrical response of the TENG is directly proportional to the voltage of the external source, as evidenced in Figure 3c,d. Conversely, the $Q_{\mathrm{SC}}$ demonstrates a decrease proportional to the square distance between the electrode and the external source, as shown in Figure 3f.

Figure 3g,h depicts the electrical characteristics of the device during operation in a single-electrode mode. In addition, Figure 3i,j provides further evidence supporting the hypothesis that TENGs can exploit both phenomena associated with displacement current, including electromagnetic induction and polarization density simultaneously [1]. Finally, it is worth noting that the electrical performance of the TENG introduced in an electric field outperforms both a basic TENG and an EFEH configuration (see Figure 3k,l).

## 4. Experimental Results

This section studies the performance of TENGs using several harvesters of varying sizes positioned within the electric field of an electric motor. In this analysis, the motor housing remains ungrounded, and the circuit for current flow is established between the harvester, load, and a grounded electrode. The TENG response is assessed based on the three specific operation modes outlined in Section 2. Lastly, potential applications are suggested, leveraging the highest-performing harvester.

### 4.1. Assessment of the TENG as EFEH

Figure 4 provides an overview of the performance of a TENG as EFEH. To isolate the harvesting process of electrostatic charges over the electric field from others derived from external motion, the single-phase electric motor was anchored to a non-vibrating mechanism that dampens vibrations and absorbs oscillations. Then, the TENG was mounted in the proximity of a motor. In addition, the mobile mechanisms of the energy harvester remained fixed during this stage. As the design, implementation, and setup of long-side harvesters could be highly difficult and impractical, we adjusted the harvesters to a maximum size of 3.5 × 10 cm^2^. The schematic of the harvesting process is shown in Figure 1b. The data presented in Figure 4a reveal an unexpected outcome in the TENG, demonstrating the ability to harvest energy from the electric field in an electrified environment. Certainly, a clear correlation is observed between the output voltage of the TENGs and EFEHs. Additionally, the output power of the TENG demonstrates dependency on the connected external load, as illustrated in Figure 4b. The peak voltage and current are illustrated in Figure 4c and Figure 4d, respectively. Similar to an EFEH by inspection it is possible to note that the output power of the TENG first increased and then decreased at a specific resistor load value, as shown in Figure 4e. Furthermore, the resistor value remained unaffected by the motor voltage and was set at 10 MΩ for the duration of this analysis. Finally, the maximum power density of the TENG working as an EFEH is 21.4 mW/m^2^ for an optimized matching state.

The experiments were conducted using a 10 MΩ resistor, hereafter referred to as the load. In Figure 4f, there is a clear trend of increasing the output power of the harvester with the increase in the electrode area. This finding reinforces the general idea that the EFEHs and TENGs depend on the common electrode area in which the electric field is induced. In addition, it is possible to note a positive correlation between the output power of the TENGs and EFEHs, as shown in Figure 4f. Moreover, following the motor’s operating voltage increase, a significant rise in the harvester power was registered. In essence, the performance of the TENG directly depends on the electric field value, and it increases as the common area mismatches the maximum electric field induction. As Figure 4g shows, the output power of the TENG significantly decreases with the distance between the electric-field radiation source and the harvester. This finding was expected since the electric field strength is generally reduced with the squared distance between the field source and the electric energy harvester. Furthermore, as presented in Figure 4h, The power density of the TENG remains consistent when the harvester is positioned at the same distance as the electric field source, irrespective of spatial orientation.

### 4.2. Performance Comparison of the Harvesting Technologies

Figure 5 presents the experimental results of the TENG in three different operation scenarios. The harvester works in single-electrode mode, i.e., the electrical motor is off, and a linear motor enables contact between mobile and fixed parts. In the following experiments, this operating mode will be referred to as single-TENG. Similar to the previous trials, here we also selected a 10 MΩ load (see Section 4.1). The solid magenta line in Figure 5b presents the output voltage of the harvester when the contact frequency is approximately 10 Hz. It is worth noting that the output voltage has two components related to mechanical stimulus and electrostatic induction. The first component originates from the contact electrification phenomenon and the triboelectric effect. On the other hand, although the motor is off, it is possible to observe that there is voltage in the test load, as shown in Table 2. Therefore, the second component is associated with electromagnetic pollution. As shown in Table 2, the results indicate that the output power of the single-TENG depends on the electrode size. Additionally, it was observed that the power extraction capability of a single TENG is significantly lower than that of the other comparison technologies presented here, as illustrated in Figure 5c. Nevertheless, the experiments can be deemed reasonably successful.

The second mode, explored in Section 4.1, will be referred hereafter to as the EFEH. The solid black line in Figure 5b depicts the output voltage of the harvester for two different operating voltages of the electric motor: 50 V and 230 V. In this test, the field source was able to produce an induced voltage in the harvester at the same frequency of 50 Hz (main frequency). In this analysis, most of the output power is generated by the electric field energy harvesting process, as detailed in Figure 5c.

Regarding the next operating mode, we studied the harvester response by merging both phenomena, i.e., (i) the electric field induction and (ii) the triboelectric effect. This hybrid case is called the EF-TENG mode. In Figure 5b, the solid black line denotes the output voltage of the EF-TENG. The most promising aspect of this result is that the induced voltage can be considered a linear combination of the voltage contributed by each technology, as shown in Table 2. As shown in Figure 5c, the instantaneous power achieves the maximum peak at a nominal source voltage for the EF-TENG mode. However, this outcome is considered marginal at best since the electrical power decreases along the remaining operating conditions. On the other hand, by the inspection of Table 2 and Figure 5d, the results show that the output power of the energy harvester increases with the harvesting plate size. This finding was expected since the output power of both technologies (EFEH and TENG) depends on the electrode size. As electric motors do not necessarily require a ground connection to operate, it was hard to close the current loop to set the harvester operation. For instance, in [44], the authors proposed to close the current loop using several surfaces as reference. Hence, we probed the harvester performance of the EF-TENG using concrete surfaces as a ground connection. In Figure 5e, there is a clear trend of an increase in the output power when the harvester works as an EF-TENG. On average, the increment of the output power of the EF-TENG was about 5%. Because the conventional external loads are low-power portable electronics, we studied the performance of the EF-TENG using a conditioned circuit as load (see Section 2.1). The conditioned circuit is composed of a full-bridge rectifier with a parallel-connected capacitor. The experimental results are shown in Figure 5f. The most relevant findings from this experiment are the following: (i) the proposed harvester can collect enough energy to drive low-power electronics; (ii) the output voltage increases in accordance with the value of the storage capacitor; (iii) the selection of the capacitor directly influences the available and collected energy. Such findings reinforce the early conclusions shown in [39].

### 4.3. Application of the Energy Harvest Technology

This work has demonstrated that TENGs can serve as an EFEH by exploiting contact electrification, the triboelectric effect, and electrostatic induction. Therefore, the integration of TENGs in electrified environments can be an essential alternative to power low-power sensors. This section provides a set of potential applications that TENGs can develop. The experiments in this section were developed as follows:Based on the previous results from Section 4.1 and Section 4.2, a 3.5 × 10 cm^2^ TENG was designed, assembled and tested.The TENG was mounted on a 1.5 HP, 8 A, 230 VAC, 50 Hz single-phase motor. The motor works under nominal operation conditions and no-load conditions.A management electric circuit consists of a full-bridge rectifier and a storage capacitor, described in Section 2.1. Although the power density of the harvester depends on the capacitor value, we selected a 4.7 μF test capacitor based on the power losses experimented in trials from Section 4.1 and Section 4.2. On the other hand, a 100 μF electrolytic capacitor was used for the following experiments because of the low operating voltage of the connected loads.According to the analysis presented in Section 2.1, the voltage profile of the load is obtained during a 15-min test using a high-precision digital electrometer.

Figure 6a show a 3.5 × 10 cm^2^ EF-TENG mounted on a single-phase motor. We used a commercial green LED operating with a nominal voltage of 3.3 V as the test load. It was possible to observe that the device operating in EF-TENG mode outperforms the EFEH mode in the collection of electrostatic charges. Thus, the approached results revealed that TENGs are emerging devices that can be used to exploit alternating electric fields and mechanical stimuli. The increase in the number of LEDs was not possible in this mode because of the low power density of the TENG. Therefore, a low-frequency duty cycle signal was applied. In this case, the transferred energy between the harvester and LEDs was performed by a DB3 Diac with a break-over voltage of 32 V. Using this adjustment, it was possible to light about five commercial green LEDs for 1.5 min when the motor operated at nominal conditions, as shown in Figure 6b. On the other hand, the harvested energy can provide power to ultra-low-power microchips, which could be used for IoT devices. As shown in Figure 6c, a 100 μW load was activated. After approximately 175 s, an external switch was remotely turned on when the interface capacitor had been charged up to 1.5 V average. The EF-TENG device provides 7 s of autonomous operation to the portable device. Since the electronic device operates over a voltage range of 0.9 to 1.5 V, the external switch is manually turned off when the voltage is less than 0.9 V. The load can be remotely turned on after 75 s. Following the former idea, a 1 mW load is driven using the proposed TENG. The TENG was mounted on the cover of two different motors: 1.25 HP, 120 V, 60 Hz, 8 A and 1.5 HP, 230 V, 50 Hz, 8 A. The electrometer is activated every 410 s to measure the surrounding electric field of a 1.25 HP motor. The experiment was repeated with a 1.5 HP motor, where the device can be turned on at 380 s. Finally, the results consolidate that the performance of the harvester depends on the operating voltage of the motor and not other electrical parameters.

## 5. Discussion

Self-powered IoE sensors aim to assist operators in enhancing the efficient management of industry assets. In recent years, this research field has seen significant advances in monitoring the health and quality of different industry mechanisms (e.g., electric machinery, generators, and electric motors) through WSNs [45]. Triboelectric nanogenerators have demonstrated promising outcomes in harvesting electrostatic charges within electrified environments, as illustrated in Table 2 and Figure 7. Four TENGs of varying sizes were evaluated, revealing superior charge collection capabilities compared to EFEHs. In particular, the 10 × 3.5 cm^2^ TENG presented better performance, achieving a notable 10.49% increase (at 50 V) in harvester power compared to EFEHs. As shown in Figure 7, the observed lower power density of the TENG mode compared to the EFEH mode can be attributed to the utilization of low-cost materials (PLA and CA) in the construction of the harvester. Present efforts are directed toward fabricating TENGs using advanced biocompatible and eco-friendly materials [2,39,46]. Many of these studies have primarily focused on substituting dielectric and metal layers with advanced materials such as aerogels or bioplastics [47,48], highlighting how the robustness and durability of the TENG is able to change the performance of the TENGs. In particular, TENGs built upon biopolymers enabled more efficient harvesting capabilities due to their covalent chemistry which enrolls research in other triboelectric materials.

In general, the decrease in the voltage of the external source harms the performance of TENGs, as illustrated in Figure 7c,g,k. Empirical evidence suggests that this phenomenon may be attributed to the similar structural characteristics of both harvesters [1,2]. Therefore, TENGs could serve as promising alternatives to EFEHs in electrified environments such as power systems. These findings contribute significantly to the existing literature, which primarily focuses on developing EFEHs for collecting electrostatic charges in electrified environments. However, experimental findings have revealed that the power density significantly decreases without control over the impact frequency in certain instances, as shown in Figure 7h. It is, therefore, advisable to analyze the frequency impact across different values to thoroughly assess its influence on the harvester’s power density, aiming to devise practical industrial application devices. Additionally, there is an evident necessity to investigate novel smart materials capable of adapting to the environment and reacting to external stimuli [49].

## 6. Conclusions

This work presented a comprehensive analysis of Tribo-Electric Nanogenerators (TENGs) as potential replacements for electric field energy harvesting in electrified environments. Findings suggest that a TENG has a similar response to EFEHs because of the identical design. In addition, the experiments indicate that TENGs could collect electrostatic charges related to a time-varying electric field. Also, one of the more significant findings to emerge from this work is that the power density of the TENG increases when the TENG experiments both mechanical stimulus and electric field. In general, this work might expand the development of self-powered technologies to monitor variables in power system scenarios properly. Experimental results showed that a 3.5 × 10 cm^2^ TENG is able to provide about 2.14 μW/cm^2^. These results were obtained when the TENG operated as an EFEH mounted on a single-phase motor working in nominal conditions. The experimentation results indicate that the electric power is enough to power commercial LEDs or drive low-power chipsets. In particular, the harvester presented in this work is capable of handling about five green LEDs each 1.5 min, a 100 μW load for 7 s in 75 s intervals or 1 mW load for 3 s in 380 s intervals.

This research work provides insights into the electric-field energy harvesting process using TENGs. Although the acquired data revealed that the TENG is a vital technology for harvesting electrostatic charges, several open issues require further study. It might be possible to determine the optimal contact frequency of the TENG and analyze the effect on the power density. In addition, a detailed study of materials is required to characterize the TENG completely within an electrified environment. An important issue for future research is also to set a functional relationship between the electrical outputs of the harvesters and the electric environmental parameters. It is relevant to ensure uniform contact between fixed and mobile parts to reduce potential measurement issues. Finally, it is recommended to maintain the temperature and humidity levels to guarantee the correct measuring and data reproducibility.

## Figures and Tables

**Figure 1 sensors-24-02507-f001:**
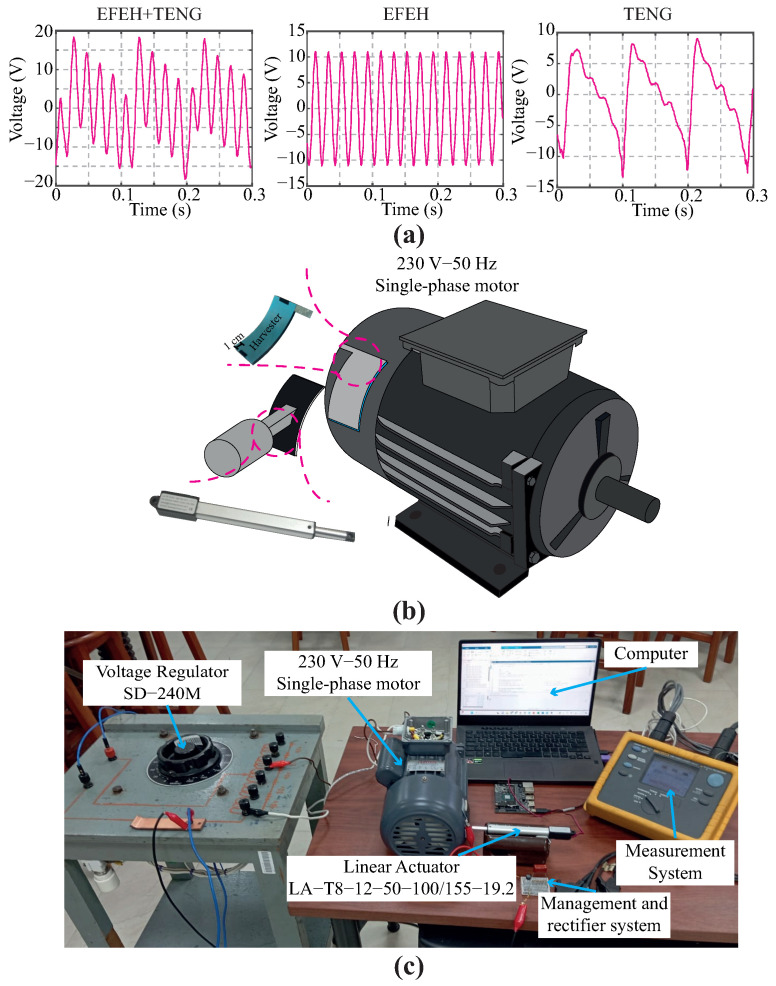
Schematic illustration of the analysis setup. (**a**) Comparison of induced voltage using TENG in the three operation modes; (**b**) Graphical representation of physical energy harvesting devices and (**c**) Experimental setup.

**Figure 2 sensors-24-02507-f002:**
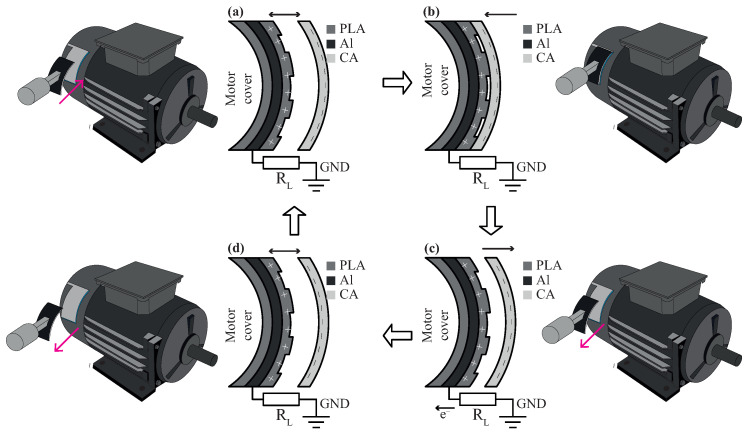
Schematic diagram of a TENG mounted on a single-phase electric motor, working in single-electrode mode. (**a**) Before contact; (**b**) Fully-contact, electric charges are generated on PVA and CA layers; (**c**) Release, electrons flow; and (**d**) Fully separate and reach an electrical equilibrium, no current flows at that moment.

**Figure 3 sensors-24-02507-f003:**
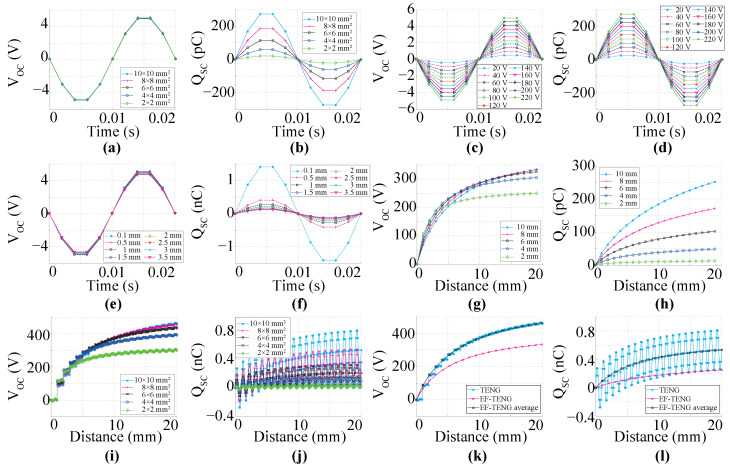
Simulation results. (**a**,**b**) The relation between the electrode area and the electrical performance of the TENG operating as an EFEH, respectively, $V_{\mathrm{OC}}$ and $Q_{\mathrm{SC}}$. (**c**,**d**) The correlation between the voltage of an external source and the electrical performance of the TENG functioning as an EFEH. (**e**,**f**) The relationship between the harvester’s location and the electrical performance of the TENG operating as an EFEH. (**g**,**h**) The correlation between the electrode area and the electrical performance of the TENG without an external voltage source. (**i**,**j**) The correlation between the electrode area and the electrical performance of the TENG operating as an EF-TENG. (**k**,**l**) Comparison of the TENG’s electrical performance across three distinct operation modes.

**Figure 4 sensors-24-02507-f004:**
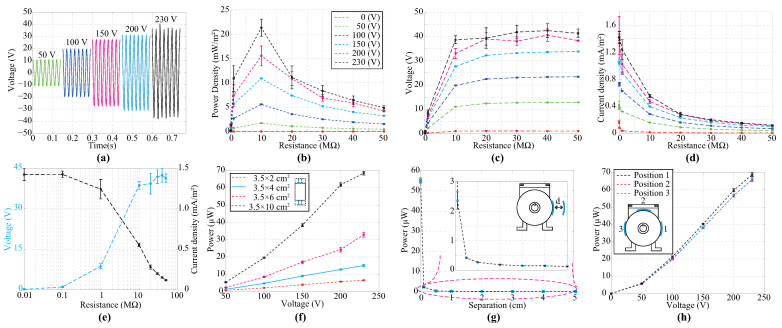
(**a**) The output voltage of the TENG regarding the electric motor voltage using a 10 MΩ resistor. (**b**) Output performance under different external loads of the proposed TENG. (**c**) The voltage of the TENG under different changes in the external source voltage across different load resistances. (**d**) The current of the TENG under different changes in the external source voltage across different load resistances. (**e**) Output performance of the proposed TENG under various external loads (in an electric field of 230 V). (**f**) The relation between the electrode area and the output power of the TENG; (**g**,**h**) The relation between the output power of TENG and the harvester location. Note: we used a 10 MΩ resistor as load in experiments (**f**–**h**).

**Figure 5 sensors-24-02507-f005:**
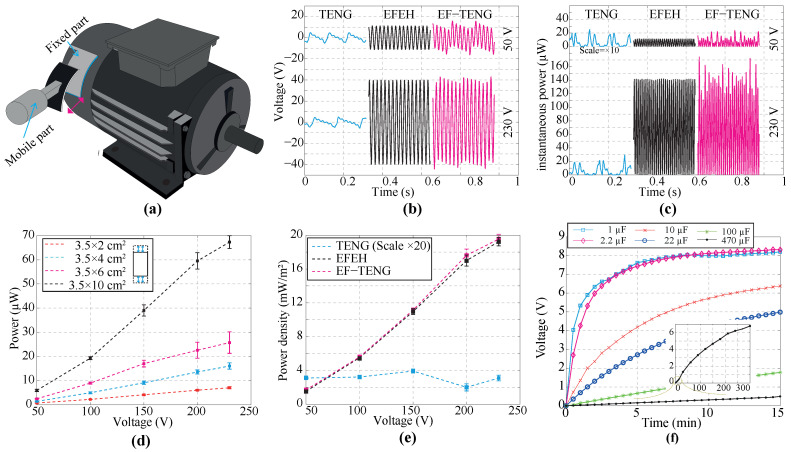
(**a**) Representative depiction of the TENG used to harvest energy from electric motors; (**b**) Comparison of the induced voltage using the TENG in the three operating modes; (**c**) Comparison of the Instantaneous power using the TENG in the three operating modes; (**d**) The relation between the electrode area and the output power of the EF-TENG; (**e**) Comparison of the output power using the TENG in the three operating modes; (**f**) Storage capacitor voltage in different commercial capacitors using a TENG working in the EF-TENG mode. Note: Two operating voltages of the electrical motor were compared in (**b**,**c**): 50 V and 230 V.

**Figure 6 sensors-24-02507-f006:**
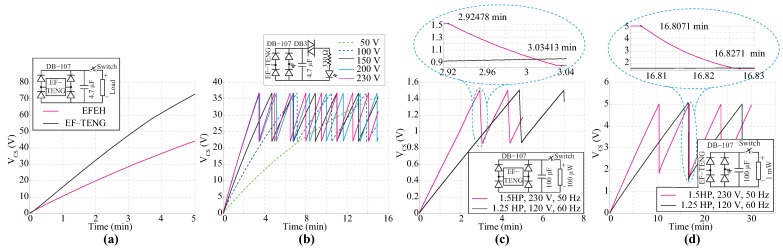
Actual applications of TENGs to harvest the energy of the surrounding electric field and mechanical stimulus in energized environments. (**a**) Comparison between an EFEH and an EF-TENG used to power portable electronics; (**b**) Five commercial LEDs are driven by an EF-TENG in a variable electric field ambient; (**c**) An EF-TENG driving a 100 μW load. (**d**) An EF-TENG driving a 1 mW load.

**Figure 7 sensors-24-02507-f007:**
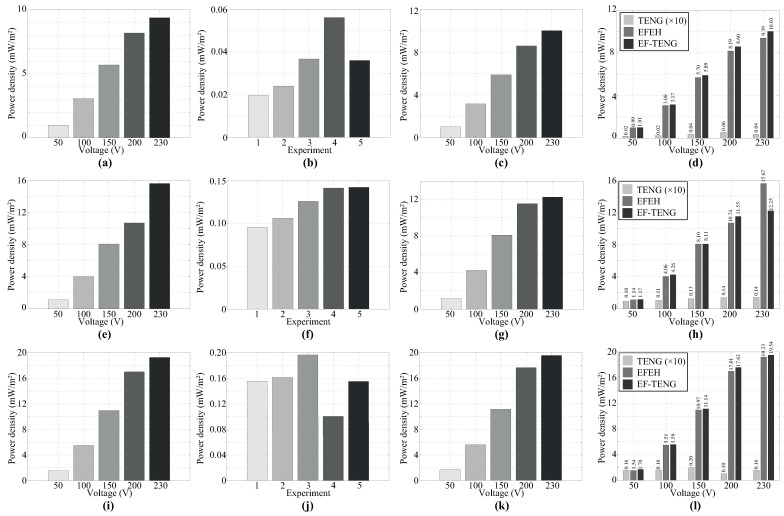
The power density of the EF-TENG is investigated across three operation modes: EFEH, TENG, and hybrid mode, with a comparison between technologies for TENG sizes. (**a**–**d**) 2 × 3.5 cm^2^; (**e**–**h**) 6 × 3.5 cm^2^, and (**i**–**l**) 10 × 3.5 cm^2^.

**Table 1 sensors-24-02507-t001:** Technical specifications of sensors and instruments used in this work.

Sensor/Instrument	Technical Specifications
Single-phase motor Imatesa, Santiago de Chile, Chile	Nominal Power: 1.10 kW Nominal speed: 2750 RPM Voltage supply: 230 V Operation frequency: 50 Hz
Creality Ender-Pro 3 3D printer Creality, Valparaíso, Chile	Printing size: 220 × 220 × 250 mm Filament: PLA Layer Thickness: 0.1–0.4 mm Print Precision: ±0.1 mm
Linear Actuator LA-T8-12-50-100/155-19.2 GoMotorWorld, Quito, Ecuador	Nominal voltage: 12 V Unload speed: 50 mm/s Load: 19.2 N Duty cycle: 10%
Voltage Regulator SD-240M Variac, Vaparaíso, Chile	Input voltage: 110–220 V Output voltage: 0–240 V Frequency: 50 Hz Maximum Power: 2 kVA
Management and rectifier circuit	−DB-107 full-bridge rectifier Comchip, Valparaíso, Chile Reverse voltage: 1000 V Voltage drop: 1.1 V Junction capacitor: 25 pF −4.7 μF metalized polypropylene film capacitor Tinyair, Santiago de Chile, Chile Leakage current: 22 nF Maximum voltage: 103 V

**Table 2 sensors-24-02507-t002:** Performance comparison of the TENG working in three operation modes: (i) as an EFEH; (ii) as a single-TENG and (iii) as an EF-TENG. All experiments were conducted under the following conditions: 10 MΩ resistor was employed as the load in all trials. * Power measurements in the single-TENG mode were obtained with the electrical motor turned off. ** The inclusion of the EFEH power acquired with a turned-off motor (0 V) is aimed at studying the effect of other external electrified sources (such as electrical wires, lights, electrical appliances, etc.) and how they are mitigated on the experiments.

Voltage (V)	Size Plate (cm × cm − cm^2^)	EFEH	TENG *	EF-TENG	Increase
		Power (μW)	Power (μW)	Power (μW)	%
0 **	10 × 3.5	0.4986 ± 0.0649	-	-	-
50	2 × 3.5	0.6919 ± 0.0227	0.0138 ± 0.039	0.7065 ± 0.0377	2.11
	4 × 3.5	1.3222 ± 0.0470	0.1565 ± 0.1412	1.4037 ± 0.1289	6.16
	6 × 3.5	2.3901 ± 0.0729	0.2010 ± 0.1533	2.4581 ± 0.1709	2.85
	10 × 3.5	5.3835 ± 0.2130	0.5434 ± 0.4633	5.9483 ± 0.4813	10.49
100	2 × 3.5	2.1529 ± 0.4881	0.0168 ± 0.0063	2.2155 ± 0.0792	2.91
	4 × 3.5	4.7411 ± 0.1244	0.2537 ± 0.2083	4.9630 ± 0.4060	4.68
	6 × 3.5	8.5167 ± 0.2768	0.2234 ± 0.2369	8.9482 ± 0.3978	5.07
	10 × 3.5	19.2855 ± 0.5903	0.5643 ± 0.5916	19.5761 ± 0.3576	1.51
150	2 × 3.5	3.9898 ± 0.0843	0.0257 ± 0.0174	4.1245 ± 0.2507	3.38
	4 × 3.5	9.0144 ± 0.1959	0.1868 ± 0.1184	9.0826 ± 0.6885	0.76
	6 × 3.5	17.0247 ± 0.6147	0.2652 ± 0.1772	17.0265 ± 1.3713	0.01
	10 × 3.5	38.3907 ± 0.8458	0.6876 ± 0.4987	39.0160 ± 2.3231	1.63
200	2 × 3.5	5.7326 ± 0.1220	0.0393 ± 0.0269	6.0181 ± 0.2719	4.98
	4 × 3.5	12.7761 ± 0.3057	0.1589 ± 0.1189	13.5376 ± 0.8585	5.96
	6 × 3.5	22.5497 ± 3.3033	0.2975 ± 0.2122	24.2505 ± 1.3204	7.54
	10 × 3.5	59.5426 ± 3.4157	0.3518 ± 0.3100	61.6962 ± 1.1257	3.62
230	2 × 3.5	6.5748 ± 0.1647	0.0252 ± 0.0204	7.0184 ± 0.4396	6.75
	4 × 3.5	15.1029 ± 0.5860	0.3291 ± 0.3181	16.0263 ± 1.2914	6.11
	6 × 3.5	32.9159 ± 1.37522	0.2996 ± 0.3374	25.7248 ± 4.450	−21.85
	10 × 3.5	67.3201 ± 2.6583	0.5418 ± 0.5484	68.4114 ± 1.1034	1.62

## Data Availability

Data are contained within the article.

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
