# Peer review of "Assessment of Triboelectric Nanogenerators for Electric Field Energy Harvesting"

_sensors, 2024, doi:10.3390/s24082507_

Round 1

Reviewer 1 Report

Comments and Suggestions for Authors

This work comprehensively evaluates the performance of TENGs in electric field energy harvesting applications. It has important research significance for expanding the application of self-powered technologies and better monitoring the variables in power system scenarios. In addition, the authors had clear and unique insights and comprehensively analyzed the current problems existing in the energy harvesting process of TENGs. However, there are still some problems that need to be solved in the article, and the following suggestions are for reference:

1. The choice of triboelectric materials is crucial to the energy harvesting efficiency of TENGs. However, this was not mentioned in this study. The introduction should be supplemented with a description of triboelectric materials and a brief explanation of how advanced triboelectric materials can improve harvesting efficiency. The following papers are recent reports on advanced triboelectric materials for the authors' reference:

â‘ Multiscale Structural Triboelectric Aerogels Enabled by Self-Assembly Driven Supramolecular Winding. Advanced Functional Materials. 2024.2400476. â‘¡ A Tough Monolithic Integrated Triboelectric Bioplastic Enabled by Dynamic Covalent Chemistry. Advanced Materials. 2024.2311993.

2. The authors briefly described the operating modes of TENGs in the study, which is not conducive to readers' clear understanding of the TENG operating mechanism. The authors need to add a concise and clear mechanism diagram to help readers understand the research mechanism more intuitively.

3. To demonstrate the comprehensiveness of the research background on the application of TENGs in harsh environments such as industry, it is suggested to refer to the following literatures, such as:â‘ Hierarchical Porous Triboelectric Aerogels Enabled by Heterointerface Engineering.  Nano Energy. 2024.121.109223.

4. Figure 4 discusses the output power of TENGs. It is worth noting that the power unit in the figure is "μW", which is not clear enough to show the power. The power density of TENG "μW/m2" should be calculated to show the power density of the material more intuitively. The voltage and current should be changed with the load, and the power calculation formula should be indicated in the study.

5. How do the authors balance the difference between the peak power and the actual load power? The authors did not show this in the experimental demonstration of the application.

6. The article conducted a large number of comparative experiments. It is necessary to add comparative data graphs of important experiments to show the differences more clearly.

Comments on the Quality of English Language

English Language should be improved

Reviewer 2 Report

Comments and Suggestions for Authors

1.       The author needs to edit Fig. 1. The Figure caption needs to be detailed and clear.

2.       Why is the voltage of the external source in Fig. 3 different from Fig. 4 and other results?

3.       The time in Fig. 5f needs to expand?

4.       Why is there a difference between 50 V (3 steps on 0.3 s) and 230 V (2 steps on 0.3 s) of TENG in Fig. 5a?

Reviewer 3 Report

Comments and Suggestions for Authors

In the manuscript, the authors reported a hybridized energy harvester composed of EFEH and TENG. We have the following comments:

1. Is there any evidence for the polarity of the surface charges? Why CA is positively charged and PLA is negatively charged? Any references or data?

2. PTFE and Nylon tribo pairs may produce larger output.

3. The legends in Fig. 2a,b, and 2i,j should be corrected to the electrode area.

4. In Table 2, why do the authors only present the EFEH power when the voltage is 0 V?

5. Could the authors give some comments on the practical application scenario of this EF-TENG? 

6. The output of TENG and EFEH is in the AC format. The induced charges on the Al electrode may be offset in some cases. Considering this condition, incorporating TENG working mode may cause a negative influence on the output of EFEH.

Comments on the Quality of English Language

Too many typos exist in the manuscript, please check it carefully.

line 52, Z. Yun should be Z. Yuan

The abbreviation of cellulose acetate is CA, not PVA. PVA usually represents Polyvinyl alcohol.

Round 2

Reviewer 1 Report

Comments and Suggestions for Authors

The revised manuscript is good for acceptance.

Reviewer 3 Report

Comments and Suggestions for Authors

We are satisfied with the response.